# Feasibility and Preliminary Efficacy of Co-Designed and Co-Created Healthy Lifestyle Social Media Intervention Programme the Daily Health Coach for Young Women: A Pilot Randomised Controlled Trial

**DOI:** 10.3390/nu16244364

**Published:** 2024-12-18

**Authors:** Jessica A. Malloy, Stephanie R. Partridge, Joya A. Kemper, Andrea Braakhuis, Rajshri Roy

**Affiliations:** 1Discipline of Nutrition and Dietetics, Faculty of Medical and Health Sciences, University of Auckland, 85 Park Road Grafton, Auckland 1011, New Zealanda.braakhuis@auckland.ac.nz (A.B.); 2Charles Perkins Centre, Faculty of Medicine and Heath, The University of Sydney, Sydney, NSW 2006, Australia; stephanie.partridge@sydney.edu.au; 3Susan Wakil School of Nursing and Midwifery, The University of Sydney, Sydney, NSW 2050, Australia; 4Management, Marketing, and Tourism, University of Canterbury, Christchurch 8041, New Zealand; 5Discipline of Nutrition and Dietetics, Susan Wakil School of Nursing and Midwifery, Faculty of Medicine and Health, The University of Sydney, Camperdown, NSW 2050, Australia

**Keywords:** social media intervention, behavioural health, diet, health promotion, young women

## Abstract

Background: Young women spend 50 min daily on social media (SM); thus, SM platforms are promising for health interventions. This study tested the feasibility and preliminary efficacy of the co-designed SM intervention the Daily Health Coach (DHC). The DHC is a 3-month healthy lifestyles intervention programme, targeting eating, physical activity, and social wellbeing behaviours in women aged 18–24, via the dissemination of health and nutrition content on social media platform Instagram. Methods: The programme was tested using an assessor-blinded, two-arm pilot randomised controlled trial with 46 participants over 12 weeks. Engagement was assessed via SM metrics; acceptability via post-programme questionnaires; and feasibility included retention, randomisation, recruitment, and data collection. Secondary outcomes—dietary quality, physical activity, social influence, disordered eating behaviours, body image, and digital health literacy—were assessed using validated surveys. Analyses included *t*-tests, chi-squared tests, and linear mixed models. The treatment effects were estimated by testing mean score differences from baseline to 3 months for intention-to-treat populations. Results: The DHC scored 83.6% for programme satisfaction. Over time, a significant decrease in body image disturbance was observed (*p* = 0.013). A significant group-by-time interaction for digital health literacy (*p* = 0.002) indicated increased ability to discern evidence-based nutrition information (*p* = 0.006). The waitlist control group showed increased social influence compared to the intervention group (*p* = 0.034). No other significant changes were observed. Conclusion: The DHC is a feasible and acceptable method for disseminating nutrition information. Larger studies are needed to determine efficacy.

## 1. Introduction

Digital marketing significantly influences young peoples’ food choices, often promoting unhealthy and highly processed options [1,2,3]. Young adults, particularly in New Zealand, are therefore over-exposed to nutrient-poor, energy-dense products, with healthier options like fruits and vegetables becoming less accessible due to cost or the appeal of processed alternatives [4].

This research targets New Zealand women aged 18–24, a group experiencing significant life transitions such as moving out, pursuing further education, starting careers, or beginning families [5,6]. These changes often negatively impact health behaviours, leading to increased alcohol consumption, poor eating habits, and reduced physical activity [7]. Globally, young adults consume the most energy-dense foods while being the poorest vegetable consumers, a pattern that often worsens in mid-adulthood [8,9]. Furthermore, compared to digital health research with children and adolescents, research targeting young adults is scarce [10]. This scarcity is compounded when focussing on young women and nutrition specifically. Greater intervention efforts are required to determine the utility of social media (SM) for supporting this population group.

In New Zealand, SM platforms are commonly adopted across demographics. A 2021 survey exploring the use of SM by New Zealand adults found that females were more likely to use the social platforms Facebook and Instagram, whilst males were more inclined to use Twitter and LinkedIn [11]. As expected, younger adults were larger consumers of SM than their older counterparts, with Instagram, Snapchat, and TikTok being the most popular platforms amongst those aged 18–29 years [11]. On a global stage, young New Zealanders spend more time on the internet than teenagers in all other countries excluding Denmark, Chile, and Sweden, with a screentime of 42 h per week, compared to the global average of 35 [12]. Although the OECD ‘PISA’ report (Programme for International Student Assessment) is not specific to SM, its findings highlight the potential of the digital environment as a promising youth health promotion space.

Aggressive digital marketing and SM now significantly influence the food environment [13,14]. Companies leverage platforms like Facebook, Instagram, and YouTube to target consumers using detailed demographic and psychological data, increasing susceptibility to unhealthy food advertising [15,16]. This targeted approach is particularly effective on young people, making them more vulnerable to discretionary food marketing. A recent study found that food brands commonly use interactive techniques including user-generated content, games, and apps to engage consumers, particularly young people, with unhealthy food promotions [17]. Compounding this, often untrained SM influencers can significantly impact young people’s nutrition beliefs and self-perceptions [18,19]. Concerning young women, repeated exposure to beauty standards and nutrition misinformation online can lead to body dissatisfaction, poor food relationships, and disordered eating behaviours [19,20,21,22].

On the positive side, digital platforms have facilitated cost-effective research and health interventions. Co-designed digital health interventions (DHIs) have shown potential for positive behaviour change and improved health outcomes, often surpassing traditional methods in reach and engagement [10]. A recent example is the ‘HEYMAN’ healthy lifestyle programme, a semi-digital intervention developed collaboratively with young men. The programme included a SM component, utilising Facebook as a platform for participant support as well as a website and wearable device [23]. ’HEYMAN’ was a feasible way of facilitating positive behaviour change, reporting significant intervention effects for weight, BMI, and energy-dense, nutrient-poor foods [23]. Like so many DHIs, the programme incorporated shared protective factors such as social support networks facilitated by SM [24,25,26,27]. This is aligned with the values of young people, who appear more receptive to holistic approaches, with strong emphases on mental and social health as well as physical states [28,29,30]. The health promotion landscape would likely benefit from an increased number of DHIs with a multi-factorial approach to behaviour change.

Co-design appears to be a favourable methodology in the development of DHIs, given its ability to enhance acceptability and effectiveness [10]. Including stakeholders with lived experience in all aspects of intervention development is an effective way of increasing intervention accessibility and use. This is particularly true of SM interventions, as platform algorithms, and therefore use, are subject to constant change. Co-design is recommended as best practice, not only to improve value and viability; participation in co-design phases has been shown to improve self-efficacy amongst designers, as well as those engaging with final outputs.

Whilst recent research is indicative of SM offering feasible avenues for youth health promotion, the ability of platforms to directly influence health behaviours, particularly in a nutrition context, remains uncertain, and requires further research through randomised controlled trials (RCTs) [31,32]. A systematic review of the recruitment and retention of young adults in 2021 highlighted poor reporting of feasibility measures and provided relevant benchmarks for assessing these contributors to intervention success [33]. Livingstone et al. have also published benchmarks for retention, recruitment, and participation via an RCT protocol to determine the feasibility of Veg4Me, a co-designed app for young adults to improve vegetable consumption [34]. Taken together, these insights will form evidence-based criteria for ascertaining the viability of this pilot intervention.

This research aims to explore how best to disseminate nutrition information via influencer techniques, using implementation science as a framework [35,36,37,38]. The objectives are as follows:Evaluate the feasibility of recruitment, randomisation, data collection methods, and retention of a youth co-designed health promotion programme for young women aged 18–24 years against evidence-based criteria.Estimate the treatment effects of the Daily Health Coach (DHC) on improving diet quality, physical activity, and other lifestyle, psychological, and social influence measures.

## 2. Materials and Methods

### 2.1. Study Design

The DHC study is an assessor-blinded, two-arm pilot randomised controlled trial (RCT) addressing the feasibility and preliminary efficacy of a 3-month programme delivered via Instagram. Following demographic data collection, young women were individually randomised to the Daily Health Coach intervention group (commenced DHC intervention immediately) or the waitlist control group (started DHC after a 3-month delay). The study design and conduct adhered to the guidelines as outlined by Thabane et al. [39]. The checklist is an adapted version of the Consolidated Standards of Reporting Trials (CONSORT) guidelines specifically for pilot studies [40]. The reporting of results follows that of Ashton et al., who conducted a similar trial with young men [23].

#### Ethics

This study was approved by the Human Participants Ethics Committee at the University of Auckland on 12 July 2023 for three years. Reference number: UAHPEC26195. The trial was retrospectively registered with the American Economic Association’s Registry for randomised controlled trials, number AEARCTR-0014872.

### 2.2. Intervention Development

A comprehensive overview of the co-design phase and programme development is available for further reading [41]. In brief, the DHC is a SM health promotion programme designed for young women aged 18–24 to improve eating habits, activity levels, and overall wellbeing. Developed using the Young & Well Co-operative Research Centre framework, which was adapted for general wellbeing and nutrition behaviours, the DHC incorporated the target audience into the design process to enhance effectiveness and address cohort-specific needs [39,40]. Formative research identified motivators, barriers, and preferences for content delivery via SM, involving young women (*n* = 19) across Aotearoa, New Zealand in co-design workshops [42]. These workshops explored the participants’ conceptualisation of health, examined influencer content on Instagram and TikTok, and used the micro–meso–macro model to discuss common barriers [43]. Insights were used to create the DHC Instagram profile and a 12-week content planner, with contributions from student dietitians and a recent nutrition graduate [41].

Informed by co-design insights and best practice guidelines for diet and physical activity, the DHC programme also incorporated the COM-B model, Self-Determination Theory (SDT), and evidence from successful health-related interventions for this demographic [42,43,44,45]. The result is a 12-week co-developed healthy lifestyles intervention programme delivered via Instagram. It involves the daily dissemination of co-created nutrition and health content via Instagram posts, reels, and stories. For users who elect to be messaged directly, the programme also involves check-ins and encouraging messages via the direct message function at their selected frequency over the 12-week intervention period.

### 2.3. Participants and Recruitment

#### 2.3.1. Sample Size

A key objective of pilot studies is to gain initial estimates for a sample size calculation in a future, adequately powered RCT, and thus a formal sample size calculation was not performed. A systematic review of pilot and feasibility studies identified a median total sample size of 30 in non-drug trials [44]. Therefore, we aimed to exceed this, and a recruitment target of 50 was set.

#### 2.3.2. Recruitment Strategy

Participants were recruited using targeted advertisements on SM pages (Facebook, Instagram, and LinkedIn), posted by the research team to personal and professional networks, as well as by the Faculty of Medical and Health Sciences at the University of Auckland. The research invitation ad directed prospective participants to an online survey to screen eligibility criteria. Eligible participants were emailed addressing their expression of interest with the participant information statement (PIS), which provided more in-depth information about the study and participant requirements (Table 1). Following the provision of the PIS, potential participants had the opportunity to ask questions (either via email/phone call or face-to-face meeting) before being sent a consent form to sign and return digitally.

The initial screening survey was administered at the recruitment stage via e-link to all potential participants and included questions to determine the self-reported adequacy of fruit and vegetable intake, exercise, and medical history, including existing diagnosis of an eating disorder (e.g., anorexia nervosa or bulimia). SM literacy was determined via a set of questions from the Social Media Literary Assessment Questionnaire, which included questions regarding specific SM-related knowledge, skills, and behaviours (ie., how familiar are you with Instagram as a SM platform?, how much do you know about the types of content that are commonly shared on SM (e.g., posts, reels, stories, polls, lives, etc.)?, and, how confident are you in your ability to use the different features and tools on SM (e.g., posting content, commenting, sharing)?). A total of eight SM familiarity questions were included in the screening survey.

### 2.4. Data Collection—Primary Outcomes

#### 2.4.1. Programme Component Use and Acceptability

The use of programme components has been objectively tracked, using SM metrics including (1) applause rate, the number of approval actions (e.g., likes, comments) a digital content receives; (2) live assessments of favourability and preference via Instagram poll or story question during the intervention; and (3) average engagement rate. Engagement with the intervention was measured via collection of data on post likes and comments. As the intervention was closed (i.e., run on a private Instagram account), post “views” were unable to be assessed, and participants were not able to share content seen with other individuals.

A post-programme process evaluation survey developed by the research team and informed by previous studies was administered to assess DHC intervention components [46]. Participants were asked to rank individual programme components on a 5-point Likert scale from strongly agree (=5) to strongly disagree (=1), for attractiveness (“visually appealing”), comprehension (“provided me with useful information”), usability (“easy to use/receive”), length (“I am satisfied with the length of the programme”), ability to persuade/engage (“helped me attain my goals”), ability to provide support (“was supportive in answering my queries/questions”), and overall satisfaction with the DHC.

#### 2.4.2. Feasibility of Recruitment

The eligibility screening survey distributed to all prospective participants allowed for the measurement of recruitment feasibility, via the number of young women interested versus those eligible. Benchmark criteria for recruitment are based on the findings of Whatnall et al., who performed a systematic review of the literature on the recruitment and retention of young adults in behavioural interventions targeting nutrition, physical activity, and/or obesity [33,34,47]. The feasibility benchmark for recruitment is >40% of young women screened for eligibility subsequently enrolled in the study.

#### 2.4.3. Feasibility of Retention

Retention has been defined as attendance at the 3-month follow-up measurements, where participants remain following the DHC on Instagram. A retention rate target of >80% for the intervention group at 3-month follow-up assessment has been established according to findings from two relevant systematic reviews and a scoping literature review of digital health interventions for young adults undertaken by the research team [34,48]

#### 2.4.4. Acceptability of Randomisation

Randomisation feasibility has been assessed by asking participants to rank overall satisfaction with the group allocation on a 5-point Likert scale from very satisfied (=5) to very unsatisfied (=1).

#### 2.4.5. Acceptability of Data Collection

Acceptability of data collection has been estimated from the percentage of young women who completed all objective and self-report questionnaires at baseline, mid-point, and after the intervention. Those who ceased completion will be included when calculating rates of participation. A participation rate of >70% has been set as an evidence-based benchmark for evaluating the feasibility of data collection methods [34].

### 2.5. Data Collection—Secondary Outcomes

#### 2.5.1. Preliminary Efficacy

All questionnaires used for the collection of data are validated study instruments. Changes to diet quality such as intake of fruits, vegetables, energy-dense take-away meals, SSB, water, and physical activity were measured via the Short Form Food Frequency (SF-FFQ) questionnaire and the International Physical Activity questionnaire (IPAQ). As well as exercise and nutrition habits, participants were instructed to complete surveys that evaluated social influence, body image disturbance, food relationships, and digital health literacy. In total, six validated surveys were distributed during each round to evaluate healthy behaviour change:A short-form dietary questionnaire (SF-FFQ) [49];Physical activity questionnaire (IPAQ) [50];Social influence questionnaire (SIQ) [51];Three-factor eating questionnaire-Revised 18-item (TFEQ-R18) [52];Body image disturbance questionnaire [53];eHEALS digital healthy literacy questionnaire [54].

#### 2.5.2. Survey Distribution

Each survey round involved the completion of the surveys (*n* = 6). Participants in the intervention group were invited to complete the following three rounds of surveys: one week prior to the 12-week intervention (preliminary data collection), mid intervention at 6 weeks, and one week post intervention period. Young women in the waitlist control group completed surveys at the same time as the intervention group when awaiting the programme, as well as two additional survey rounds when receiving the intervention (Figure 1). A total of 18 surveys required completion by the intervention group, and 30 for the waitlist control group.

### 2.6. Randomisation

Participants were randomised using computer generation via the Excel randomisation function by the student researcher, who utilised participant UPIs for input. The ratio of assignment to groups was 50:50. Half of the participants were randomised to the intervention group (*n* = 25), and half were allocated to the waitlist control group (*n* = 25). The research team had no say over which group participants were allocated. This study was an open-label study; participants were made aware of which group they were assigned to upon allocation.

### 2.7. Statistical Analysis

The data have been analysed using IBM SPSS Statistical software [55]. The differences between groups at baseline and the characteristics of completers vs. dropouts were tested using independent *t*-tests for continuous variables and chi-squared (χ2) tests for categorical variables. The significance level for the comparison of baseline characteristics was set at 0.05. Programme acceptability measures have been presented as mean ±SD, with higher scores (maximum of 5) indicating greater acceptability/satisfaction.

All secondary health outcomes were included in linear mixed model analyses; the predictors (fixed effects) included time (treated as categorical with levels baseline, mid-point and 3 months), treatment group (intervention and control), and an interaction term for time by treatment group. Covariate type AR(1) was selected as suitable for the longitudinal data. The *p* value associated with the interaction term was used to determine the statistical significance of any difference between treatment groups. All participants who completed at least one survey round whilst receiving the DHC intervention were included in linear mixed model analysis to assess change over time.

For the estimation of treatment effects, differences in mean scores from baseline to 3 months were tested for intention-to-treat (ITT) populations. Differences of means and 95% confidence intervals were calculated using the Cohen’s d equation for mean change from baseline within and between groups [54]. Only participants who completed baseline and end-point surveys at 3 months were included in this analysis.

## 3. Results

### 3.1. Participant Flow at Each Stage

Almost all participants remained in the DHC intervention for the 12-week research period (Figure 2). One participant withdrew from the intervention for personal reasons within the first month. Two participants from the waitlist control group never followed the DHC on Instagram for the duration of the programme. One participant in the same group unfollowed the DHC without providing a reason. Therefore, despite the recruitment of 50 participants, 46 were retained throughout the 3-month intervention period.

### 3.2. Baseline Data

There were no significant differences between intervention and control groups for any of the demographic factors assessed at baseline between groups, or between completers vs. dropouts (Table 2). All recruited participants (*n* = 50) were included in baseline analyses.

### 3.3. Primary Results

#### 3.3.1. Feasibility of Research Procedures; Programme Component Acceptability and Use

The feasibility of programme components is presented as mean scores for post-programme evaluation survey (PPE) responses from a 5-point Likert scale. The PPE was deemed as reliable following calculation of the a-Chronbach value, which was 0.7__. Overall, the DHC scored 83.6% (4.18/5) for programme satisfaction (Table 3). Participants found the DHC to be useful, aesthetically pleasing, accessible, and appreciated intervention components. Improvements to support individual queries and goals should be a key focus for future iterations.

The total rate of engagement with the DHC programme during intervention periods was 10.04%. The average engagement rate for influencers across Instagram varies by source from 1% to 3% [56,57,58]. Micro-influencers, such as the DHC, tend to see higher rates of engagement [58]. For smaller influencers, an engagement rate above 5% is said to be indicative of strong engagement [58]. The main form of engagement was via post ‘likes’. Only three users left comments on posts across both cohorts. The mean like count was 4.62, ranging from 0 to 13. Posts were more ‘liked’ than reels. The top-rated or ‘liked’ post topics for both groups were nutrition misinformation, a nutritious carbohydrate explanation, a recipe for a glazed salmon bowl, a ketogenic diet ‘myth-bust’, and nutritious pantry staple ideas.

#### 3.3.2. Feasibility of Recruitment

Recruitment feasibility was calculated as the number of valid applicants that completed the online screening survey and met participation criteria. The rate of qualified applicants for the DHC trial was 74.5% (78/106). This rate is deemed feasible against the pre-determined evidence-based benchmark criteria of 55% eligibility [33]. In total, 106 valid applicants expressed interest in the DHC trial. Reasons for exclusion included meeting the national fruit, vegetable, and physical activity guidelines (*n* = 6), being outside of the age range (*n* = 5), having an active eating disorder (*n* = 2), non-NZ residency (*n* = 1), concurrent participation in other healthy lifestyle programmes (*n* = 1), and incomplete eligibility screening (*n* = 11).

#### 3.3.3. Feasibility of Retention

The rate of retention was calculated as participants who followed the DHC on Instagram for the duration of the programme (*n* = 46). Participants were not retained if they (a) never followed the DHC on Instagram (*n* = 2), (b) unfollowed the DHC on Instagram (*n* = 1), or (c) actively withdrew (*n* = 1). The benchmark for successful retention was set as >66%, in accordance with similar app-based interventions [33,48]. The total rate of retention for the DHC feasibility trial was 92% (46/50).

#### 3.3.4. Acceptability of Randomisation

Randomisation satisfaction is presented as means from a 5-point Likert scale. Overall satisfaction with allocated research groups was 4.3/5 (86%). Of those who completed the randomisation satisfaction survey (*n* = 37), participants were more satisfied with being randomised to the waitlist control group (4.3 (86%) vs. 3.65/5 (73%)). This may be due to greater reimbursement resulting from additional survey rounds.

#### 3.3.5. Acceptability of Data Collection

The number of young women who completed all objective and self-report questionnaires at baseline, mid-point, and after the intervention determined the acceptability of the data collection methods, defined as the participation rate. This was found to be 80% for the DHC (40/50), exceeding the pre-determined target of >70% [34]. Regarding survey completion, *n* = 5 participants stopped completing questionnaires, yet remained following the DHC (*n* = 3 in the intervention group, *n* = 2 in the waitlist control group). One further participant missed a mid-intervention survey round. A number of participants completed all validated surveys, yet did not complete either one or both of the randomisation satisfaction surveys or the post-programme process evaluation survey (*n* = 13).

### 3.4. Estimation of Treatment Effects; Efficacy of Nutrition Habits, Digital Health Literacy, Food Relationships/Body Image, and Social Influence

No significant differences between groups were observed for any secondary measure when assessing mean change in score from baseline to three months (end of intervention period) (Table 4). Disordered eating behaviours decreased from baseline, as well as body image disturbance and physical activity. Digital health literacy and diet quality scores increased from baseline for both groups; however, this increase was not significant.

Linear mixed model analyses found that statistically significant changes were observed for three out of seven measures: body image disturbance, social influence, and digital health literacy (Table 5). A significant decrease in body image disturbance was observed over time (*p* = 0.01). There was a significant group-by-time interaction effect observed for digital health literacy (*p* = 0.002), indicating an increase in the cohort’s ability to source and/or discern evidence-based nutrition information over time (*p* = 0.01). Finally, an increase in social influence for the waitlist control group when compared to the intervention group was found, where the waitlist control group observed an increase in score, whilst the intervention group mean score declined (*p* = 0.03). No other significant changes were observed for the measured fixed effects across cohorts (between groups, over time, or group-by-time effects).

## 4. Discussion

### 4.1. Feasibility

The Daily Health Coach intervention is a feasible form of disseminating nutrition information to young women. Overall, participants found the Instagram programme to be accessible, usable, and visually appealing. Participant satisfaction was the lowest for query support and programme length, suggesting future iterations of the programme should consider a longer intervention period, as well as offering more frequent opportunities for participants to answer questions and gain support (i.e., increase in general query polls and/or direct messages). The feasibility of the DHC aligns with similar findings for youth digital health interventions, where co-designed digital tools are found to be acceptable, usable, and feasible by young people [26,59,60,61,62,63]. However, the translation of feasible digital tools or programmes to improvements in health outcomes or behaviours is, as one may anticipate, more difficult to achieve.

### 4.2. Efficacy

The DHC had no significant impact on diet quality, physical activity, or disordered eating behaviours such as uncontrolled and emotional eating. However, the findings suggest that the programme may have a positive impact on the body image and digital health literacy of young women. This aligns with the results of preliminary co-design work when developing the DHC, where young women acknowledged the role of body image and misinformation on nutrition habits and status and advocated for its inclusion in the programme [41]. These aspects of nutrition for young women are frequently missed out of general nutrition dialogue online, and the ‘grittiness’ of information on these pertinent factors may have played a role in the reported results.

The Instagram newsfeed is often flooded with information about nutrition, food, and physical activity. However, discussions of body image, nutrition misinformation, and the inflammation of these issues via SM are often absent. A 2016 Australian study investigated knowledge translation to “sticky” SM health messages, meaning content more likely to be recalled by consumers [64]. Potential influences on the “stickiness” of SM posts relevant to the DHC involve unexpected content, social currency, stories and emotion, and posts that were credible and held practical value [64]. Information regarding restrictive diet cycles, the perpetuation of beauty standards, and nutrition confusion are relatively novel, emotive, and relevant, versus general nutrition and physical activity information. Despite the discussed observations, the impact of the DHC on health outcomes remains to be determined in a larger RCT.

### 4.3. Comparison to Other Work

This study is the first of its kind to be conducted in Aotearoa, New Zealand. The evidence base for health promotion via SM continues to grow, yet interventions tend to be classified as health promotion campaigns, rather than programmes [65,66,67,68,69]. Facebook and Twitter appear to be the most common social networking platforms for disseminating health information [70]. A 2021 systematic review of the effect of SM interventions on physical activity and dietary behaviours in young people found positive effects of the reviewed interventions, demonstrating the promise of SM use in behaviour change; the same was found of a similar review conducted in 2018 [71,72]. A comparable study to the DHC was conducted in 2017, where a pilot RCT tested the feasibility of a 3-month healthy lifestyles programme, utilising Facebook as an intervention element. The programme, which was tested with 50 participants, was found to be feasible. Despite acceptability findings, efficacy results were mixed. This appears to be common across the DHI literature, with a 2022 study into another social media-based health intervention for young people disseminating similar ‘limited efficacy’ results, as well as a 2019 SM study targeting pregnant adolescents and adult women [73,74]. Although alike, these DHIs were not targeted specifically to young women, and included distinct evidence-based information concerning mental wellbeing and or physical activity, rather than solely nutrition education and awareness of body image and food relationships.

Specific to the intersection of body image and SM, in March 2024 a new project was launched in the UK involving the development of a toolkit to “equip young women with the skills and knowledge needed to cope with potential harmful social media content” [75]. The toolkit is being co-created by researchers at the University of Portsmouth, The Girls’ Network, and the target demographic.

Over the 12-week DHC programme, over 70 distinct topics associated with nutrition were shared with participants. The wide-reaching topics of conversation increased the likelihood that information would resonate with participants. The co-design and co-creation of content is also likely to be a contributing factor to the acceptability of the intervention, as collaborative design and user-generated content is known to increase reception and resonation, particularly for young people [76,77].

### 4.4. Limitations

SM is a difficult landscape to work with for many reasons. One pertinent issue is the importance of engagement for efficacy. Engagement with intervention postings is essential to continue seeing content; it is necessary to ‘tell’ algorithms that you are ‘interested’ in content via liking, sharing, and/or commenting. If posts are not interacted with in early intervention stages, it is unlikely that they will continue to appear on participants’ newsfeeds. To overcome this, followers of the DHC were instructed to engage with any content seen in the first week of the intervention to increase the likelihood of continual newsfeed presence. The first week of engagement metrics have therefore been removed from analyses. However, there is still a chance that pseudo-engagement has persisted, clouding results.

As with all health dissemination research undertaken on SM, it is not possible to state with certainty that the presented results are associated with programme material specifically. There is always a chance of users seeing similar content from other creators when online. This is further confounded by the likelihood of algorithms changing when users begin interacting with programme material, increasing the likelihood of being presented with similar nutrition or health information from alternative sources. SM algorithms change frequently and discreetly; it is therefore important for future iterations of the DHC and other DHIs to engage with social marketing experts in developmental stages to understand and best overcome potential platform barriers.

An oversight when converting the TFEQ-R18 to Redcap distribution software resulted in the absence of the final survey question. This meant that the insights for the ‘cognitive restraint’ section of the questionnaire could not be assessed with validity, and results were therefore excluded from the presented analyses. Results for cognitive restraint were non-significant for all fixed effects (*p* = 0.375 for group, *p* = 0.339 for time, and *p* = 0.908 for group * time).

### 4.5. Implications

The feasibility of the DHC confirms the hypothesis that influencer communication techniques can be used to disseminate evidence-based nutrition information to young people. These findings may be used to advocate for an amplified presence of health professionals on social networks. It is increasingly important for health professionals, particularly dietitians, to advocate for and voice their expertise across social platforms. This includes advice on the sourcing of evidence-based information to combat nutrition misinformation and confusion for young people. Furthermore, when discussing nutrition information online, it is important to reflect upon the impact of nutrition dialogue on body image. For example, those sharing content should consider terminology and avoid moralising language concerning food. General nutrition information that dispels common myths and promotes a non-diet approach can be helpful for vulnerable populations such as young women, for whom SM use often plays a role in poor body esteem or disturbance [78,79,80].

Regarding future research directions, the proven feasibility of the DHC programme provides a blueprint for potential youth digital interventions targeting health behaviours. The co-creation protocol, as well as the feasibility findings, may be referenced by researchers or health professionals looking to utilise SM as a platform for influencing positive behaviour change. This is particularly important in the nutrition space, as the majority of DHI research to date has been targeted at mental wellbeing and/or physical activity, rather than diet quality and body image. The feasibility of the DHC is owed to the collaborative design of the programme, whereby young women contributed to the development and creation of intervention content. As such, it is recommended that future DHIs are co-designed with target users to ensure acceptability, relevancy, and use.

## 5. Conclusions

The Daily Health Coach is a feasible health promotion intervention that uses Instagram as platform to reach young women. The current pilot study’s findings indicate that the research procedures, including recruitment, retention, randomisation, and data collection, are sufficiently feasible to warrant a full-scale RCT, with only minor adjustments needed. Acceptability findings are aligned with other digital health interventions created for and by young people. A larger RCT is needed to explore how best to translate feasible SM interventions to positive ‘off-screen’ changes in health behaviours.

## Figures and Tables

**Figure 1 nutrients-16-04364-f001:**
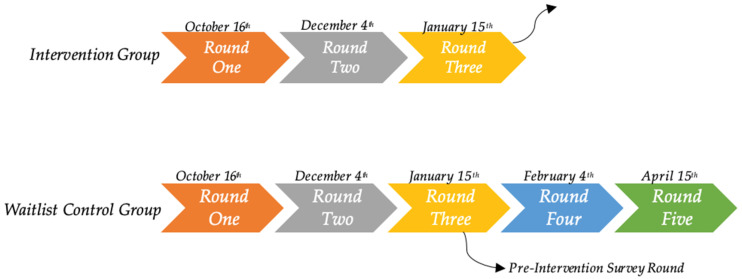
Daily Health Coach survey distribution rounds.

**Figure 2 nutrients-16-04364-f002:**
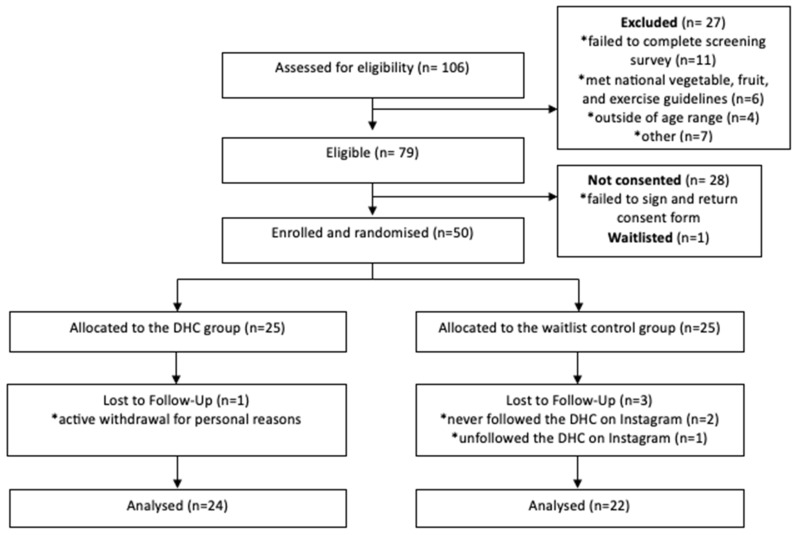
CONSORT flow chart of participant recruitment and retention throughout the DHC intervention.

**Table 1 nutrients-16-04364-t001:** Inclusion and exclusion criteria for the Daily Health Coach Trial.

Inclusion Criteria	Exclusion Criteria
Aged 18 to 24 years inclusive.	Individuals unable to give informed consent due to diminishing comprehension or understanding, and/or those with a disability (e.g., sight or hearing impairment) that precludes participation.
Identifies as female or non-binary.	Self-reported meeting national recommendations for fruit and vegetable intake (based on age/sex recommendations) ^a^ and self-reported meeting physical activity recommendations ^b^ [45].
SM literate according to study-specific criteria (outlined below).	Non-English speaking.
Available for intervention.	Currently participating in an alternative healthy lifestyle programme.
Access to a computer, tablet or smartphone with email and internet facilities.	History of major medical problems ^c^, therefore not granted GP approval to participate ^d^, and/or diagnosed with an active eating disorder.

^a^ 5 servings of vegetables and 2 servings of fruit per day. ^b^ At least 2 ½ hours of moderate (30 min/day) or 1 ¼ hours of vigorous physical activity spread throughout the week. ^c^ Including heart disease or diabetes that requires insulin injections. ^d^ Those answering ‘yes’ to any of the conditions in the pre-medial exercise screener require GP approval to participate.

**Table 2 nutrients-16-04364-t002:** Baseline demographics and characteristics of study participants.

Variable	Categories	Intervention Group (IG) *n* = 25 Mean (SD) *p*-Value	Waitlist Control Group (WCG) *n* = 25 Mean (SD) *p*-Value	Total *n* = 50 Mean (SD) *p*-Value	Completers *n* = 46 Mean (SD) *p*-Value	Dropouts *n* = 4 Mean (SD) *p*-Value
Age (years)		20.8 (2.20)	21.88 (1.79)	21.34 (2.02) 0.63 ^(a)^	21.29 (2.06) 0.509 ^(a)^	22.00 (2.16)
Ethnicity	New Zealand European	52% (13)	68% (17)	60% (30)	59% (27)	75% (3)
	Indian	0% (0)	8% (2)	4% (2)	2% (1)	25% (1)
	Chinese	16% (4)	16% (4)	16% (8)	17% (8)	0% (0)
	Other (e.g., Dutch, Japanese)	8% (2)	32% (8)	20% (10) 0.105 ^(b)^	22% (10) 0.089 ^(b)^	0% (0)
Employment Status	Currently studying/student	76% (19)	72% (18)	74% (37)	72% (33)	100% (4)
	Employed, ≥40 h/week	12% (3)	20% (5)	16% (8)	17% (8)	0% (0)
	Employed, <40 h/week	12% (3)	8% (2)	10% (5) 0.695 ^(b)^	11% (5) 0.466 ^(b)^	0% (0)
Vegetable Consumption	None	0% (0)	0% (0)	0% (0)	N/A	N/A
	One serving	20% (5)	12% (3)	16% (8)		
	Two servings	16% (4)	44% (11)	30% (15)		
	Three servings	32% (8)	20% (5)	26% (13)		
	Four servings	32% (8)	20% (5)	26% (13)		
	Five or more servings	0% (0)	4% (1)	2% (1) 0.188 ^(b)^		
Fruit Consumption	None	8% (2)	16% (4)	12% (6)	N/A	N/A
	One piece/serving	36% (9)	52% (13)	44% (22)		
	Two or more pieces/servings	56% (14)	32% (8)	44% (22) 0.220 ^(b)^		
Physical Activity	<1 h/week	8% (2)	0% (0)	4% (2)	N/A	N/A
	~1 h/week	4% (1)	8% (2)	6% (3)		
	~1.5 h/week	8% (2)	4% (1)	6% (3)		
	Up to 2 h/week	28% (7)	12% (3)	20% (10)		
	~2.5 h/week	16% (4)	16% (4)	16% (8)		
	>3 h/week	48% (12)	60% (15)	54% (27) 0.678 ^(b)^		
Social media (SM) Frequency	Never	0% (0)	0% (0)	0% (0)	N/A	N/A
	Every couple of weeks	0% (0)	4% (1)	2% (1)		
	Multiple times a day	12% (3)	0% (0)	6% (3)		
	Daily	28% (7)	32% (8)	30% (15)		
	Multiple times a day	60% (15)	64% (16)	62% (31)		
	*p*-value			0.251 ^(b)^		
SM Familiarity	Not familiar at all	0% (0)	0% (0)	0% (0)	N/A	N/A
	Slightly familiar	0% (0)	0% (0)	0% (0)		
	Moderately familiar	8% (2)	0% (0)	4% (2)		
	Very familiar	16% (4)	20% (5)	18% (9)		
	Extremely familiar	76% (19)	80% (20)	78% (39) 0.344 ^(b)^		
SM Engagement	A few times a year	4% (1)	0% (0)	2% (1)	N/A	N/A
	A few times a month	4% (1)	4% (1)	4% (2)		
	Weekly	12% (3)	0% (0)	6% (3)		
	Multiple times a week	12% (3)	8% (2)	10% (5)		
	Daily	40% (10)	44% (11)	42% (21)		
	Multiple times a day	28% (7)	44% (11)	36% (18) 0.399 ^(b)^		
SM Health-Seeking Behaviours	Never	0% (0)	4% (1)	2% (1)	N/A	N/A
	Very occasionally	44% (11)	16% (4)	26% (13)		
	Sometimes	32% (8)	24% (6)	28% (14)		
	Often	24% (6)	48% (12)	36% (18)		
	All the time	8% (2)	8% (2)	8% (4) 0.267 ^(b)^		
Baseline Energy Intake	Mean (kJ/day)	7151.59 ^(1)^	6826.50 ^(1)^	6996.43 ^(1)^ 0.565 ^(a)^	N/A	N/A
Energy Intake from Alcohol	Mean (% per day)	0.26 (0.33) ^(1)^	0.33 (0.50) ^(1)^	0.30 (0.41) ^(1)^ 0.584 ^(a)^		
Baseline Diet Quality	ARFS Score	26.09 ^(1)^	25.38 ^(1)^	25.75 ^(1)^ 0.711 ^(a)^		
	Proportion of kJ from nutrient-dense foods	56% ^(1)^	54% ^(1)^	55% ^(1)^ 0.304 ^(a)^		
	Proportion of kJ from nutrient-poor foods	24% ^(1)^	24% ^(1)^	24% ^(1)^ 0.986 ^(a)^		

N/A: Not Applicable or Data Not Available. ^(a)^ Independent sample *t*-test; ^(b)^ Pearson’s chi-squared test. ^(1)^ For these variables, the sample sizes are IG (*n* = 23), WCG (*n* = 21), and Total (*n* = 44) due to incomplete dietary data.

**Table 3 nutrients-16-04364-t003:** Post-programme acceptability evaluation scores.

Programme Measure	Component Question	IG ^a^	WCG ^b^	Total
		Mean (±SD)	Mean (±SD)	Mean (±SD)
Aesthetic	“The Daily Health Coach programme, including its content, is visually appealing.”	4.16 (±1.12)	4.45 (±0.69)	4.31 (±0.92)
Usefulness	“The Daily Health Coach programme provided me with useful information, which was easily understood.”	4.47 (±0.77)	4.5 (±0.61)	4.49 (±0.68)
Accessibility	“The Daily Health Coach programme was easy to use and information was easy to find and receive.”	4.63 (±0.50)	4.15 (±0.93)	4.38 (±0.78)
Query Support	“The Daily Health Coach team were supportive in answering my queries and questions.”	4.11 (±0.94)	4.2 (±0.89)	4.15 (±0.90)
Length	“I am satisfied with the length of the Daily Health Coach programme (12 weeks).”	4.21 (±0.92)	4.2 (±0.95)	4.21 (±0.92)
Component Satisfaction	“I am satisfied with individual components such as Instagram stories, Instagram posts, Instagram/TikTok reels and Direct Messaging.”	4.21 (±0.71)	4.45 (±0.76)	4.33 (±0.74)
Goal Support	“Upon reflection, I believe I have achieved my initial goal(s) or intention(s) for the Daily Health Coach intervention.”	3.47 (±1.17)	3.75 (±0.79)	3.62 (±0.99)
Overall Satisfaction	“Overall, I am satisfied with the Daily Health Coach programme.”	4.16 (±0.96)	4.2 (±0.77)	4.18 (±0.85)

^a^ Intervention group; ^b^ waitlist control group.

**Table 4 nutrients-16-04364-t004:** Mean change in outcome within groups and differences between groups (intention-to-treat populations) at 3 months.

	Mean Change from Baseline (SD)				
Outcomes	Intervention Group	Waitlist Control Group	Mean Difference Between Groups (95% CI)	*p*-Value	Effect Size (Cohen’s d)
Uncontrolled eating (total score, *n* = 41)	−5.11 (15.91)	−7.04 (11.95)	1.92 (−7.00, 10.85)	0.67	0.14
Emotional eating (total score, *n* = 41)	−7.41 (18.70)	−2.78 (20.03)	−4.63 (−16.87, 7.61)	0.45	−0.24
Body image disturbance (total score, *n* = 41)	−0.62 (5.23)	−1.50 (3.72)	0.88 (−2.00, 3.76)	0.54	0.19
Social influence (total score, *n* = 41)	−0.95 (14.99)	5.60 (14.37)	−6.55 (−15.84, 2.73)	0.16	−0.45
Physical activity (METs/week, *n* = 21)	−40.67 (1452.79)	−419.03 (1984.88)	−221.63 (−2101.81, 1658.55)	0.81	−0.12
Digital health literacy (total score, *n* = 41)	3.52 (4.70)	2.40 (5.29)	1.22 (−2.03, 4.28)	0.48	0.23
Diet quality (ARFS score, *n* = 41)	1.71 (7.18)	0.40 (5.53)	1.31 (−2.75, 5.38)	0.52	0.20

**Table 5 nutrients-16-04364-t005:** Linear mixed model outcomes for intervention group and time in intervention.

	Type III Tests of Fixed Effects (Sig.)		
Outcomes	Group	Time	Group × Time
Uncontrolled eating (*n* = 42)	0.70	0.08	0.73
Emotional eating (*n* = 42)	0.71	0.16	0.75
Body image disturbance (*n* = 42)	0.71	0.01	0.39
Social influence (*n* = 42)	0.03	0.13	0.19
Physical activity (METs/week) (*n* = 28)	0.32	0.51	0.91
Digital health literacy (*n* = 42)	0.67	0.01	0.002
Diet quality (*n* = 42)	0.25	0.47	0.73

## Data Availability

The data presented in this study are available on request from the corresponding author.

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
