# Peer review of "Feasibility and Preliminary Efficacy of Co-Designed and Co-Created Healthy Lifestyle Social Media Intervention Programme the Daily Health Coach for Young Women: A Pilot Randomised Controlled Trial"

_nutrients, 2024, doi:10.3390/nu16244364_

Round 1

Reviewer 1 Report

Comments and Suggestions for Authors

My recommendations are the following:

Abstract: I recommend that DHC be described descriptively.

Lines 26-27 I recommend moving to the Method section of the abstract not to Results.

Lines 86 and 88 are some unspecified signs, I recommend correcting them.

Line 87 I recommend mentioning the bibliographic index number for Livingstone et al.

According to subsection 3.1. the final number of subjects in both groups is 46, I recommend clarifications. According to the other subsections it is mentioned that the total number is 50. I recommend that in the Abstract section the final number of subjects be mentioned, i.e. 46 and not 50, due to dropouts.

Table 2 I recommend that in the first line where the indicators are presented, what the presented values ​​represent should be mentioned.

Subsection 3.3.1. I recommend mentioning the reliability of the questionnaire by calculating the α-Cronhbach value. Table 3 recommends that all acronyms be descriptively mentioned below the table.

Table 4 recommends that the first line should mention what the values ​​mentioned in parentheses represent.

I recommend expanding the Discussions section by making new comparisons between the results of this study with results from previous studies.

I recommend mentioning future research directions.

In conclusion, I recommend a reorganization of section 2. Materials and Methods, presenting numerous subsections, for example: in the Participants section I recommend incorporating the Sample Size subsection and subsection 2.8., without duplicating information.

Ethical aspects in the Study design subsection.

Author Response

Please find attached responses to reviewer comments.

Reviewer 2 Report

Comments and Suggestions for Authors

paper - Feasibility and preliminary efficacy of a co-designed and co-created healthy lifestyle social media intervention program 

the Daily Health Coach’for young women: a pilot randomised controlled trial. 

here are some comments, questions, and recommendations

in the abstract - DHC - should provide the entire text on first occurence "Daily Health Coach"

also note why health interventions are needed for women aged 18-24

with the introduction - are there some statistics (percentages) comparing global and New Zealanders?

similar with the social media usage - are there some statistics?

literature seems fine

method adequate

while the statistics are not hard to understand, most discussions are needed for the different intervention group

why 18 to 24 years? 

results and discussions are appropriate 

would recommend to provide additional more practical implications - what now?

Author Response

Please find attached responses to reviewer's comments.

Round 2

Reviewer 1 Report

Comments and Suggestions for Authors

No comments